# Metabolic Pathway Reconstruction Indicates the Presence of Important Medicinal Compounds in *Coffea* Such as L-DOPA

**DOI:** 10.3390/ijms241512466

**Published:** 2023-08-05

**Authors:** Thales Henrique Cherubino Ribeiro, Raphael Ricon de Oliveira, Taís Teixeira das Neves, Wilder Douglas Santiago, Bethania Leite Mansur, Adelir Aparecida Saczk, Mario Lucio Vilela de Resende, Antonio Chalfun-Junior

**Affiliations:** 1Laboratory of Plant Molecular Physiology, Plant Physiology Sector, Department of Biology, Federal University of Lavras (UFLA), Lavras 37200-000, Brazil; tcherubino@danforthcenter.org (T.H.C.R.); rapharicon@gmail.com (R.R.d.O.); 2Plant Physiology Sector, Department of Biology, Federal University of Lavras (UFLA), Lavras 37200-000, Brazil; taisneves@ufla.br; 3National Institute of Coffee Science and Technology (INCT-CAFÉ), Federal University of Lavras (UFLA), Lavras 37200-000, Brazil; wilderdsantiago@gmail.com; 4Multiuser Instrumental Analysis Laboratory (LabMAI), Federal University of Lavras (UFLA), Lavras 37200-000, Brazil; bethania.mansur1@estudante.ufla.br; 5Analytical and Electroanalytical Laboratory (LAE), Federal University of Lavras (UFLA), Lavras 37200-000, Brazil; adelir@ufla.br; 6Plant Pathology Department, Federal University of Lavras (UFLA), Lavras 37200-000, Brazil; mlucio@ufla.br

**Keywords:** *Coffea*, DOPA DECARBOXYLASES (DDCs), L-DOPA, POLYPHENOL OXIDASES (PPOs)

## Abstract

The use of transcriptomic data to make inferences about plant metabolomes is a useful tool to help the discovery of important compounds in the available biodiversity. To unveil previously undiscovered metabolites of *Coffea*, of phytotherapeutic and economic value, we employed 24 RNAseq libraries. These libraries were sequenced from leaves exposed to a diverse range of environmental conditions. Subsequently, the data were meticulously processed to create models of putative metabolic networks, which shed light on the production of potential natural compounds of significant interest. Then, we selected one of the predicted compounds, the L-3,4-dihydroxyphenylalanine (L-DOPA), to be analyzed by LC–MS/MS using three biological replicates of flowers, leaves, and fruits from *Coffea arabica* and *Coffea canephora.* We were able to identify metabolic pathways responsible for producing several compounds of economic importance. One of the identified pathways involved in isoquinoline alkaloid biosynthesis was found to be active and producing L-DOPA, which is a common product of POLYPHENOL OXIDASES (PPOs, EC 1.14.18.1 and EC 1.10.3.1). We show that coffee plants are a natural source of L-DOPA, a widely used medicine for treatment of the human neurodegenerative condition called Parkinson’s disease. In addition, dozens of other compounds with medicinal significance were predicted as potential natural coffee products. By further refining analytical chemistry techniques, it will be possible to enhance the characterization of coffee metabolites, enabling a deeper understanding of their properties and potential applications in medicine.

## 1. Introduction

Coffee (Rubiaceae) is an important crop in which beans are harvested and roasted before being traded as a commodity [1]. It is produced mostly in tropical countries and is an important source of livelihood for millions of smallholder farmers and workers involved in the various steps of the coffee bean processing and trade [2]. *Coffea arabica* L. is the main source of coffee beans. This species is an interspecific hybrid of *Coffea canephora* Pierre and *Coffea eugenioides* Moore ancestors [3]. Among those parental species, only *C. canephora* is economically cultivated. The polyploidy of *C. arabica* (2*n* = 44) may provide physiological advantages to cope with abiotic stresses, improve phenotypic homeostasis [4], and allow a broader diversification of metabolite compounds—when compared to progenitor species—through the differential expression of homoeologous genes [5].

Beyond its beans, coffee leaves possess the potential to serve as a valuable source of economically important metabolites [6,7,8,9,10]. The tea derived from coffee leaves is abundant in natural polyphenolic compounds such as chlorogenic acids and xanthones, which act as vital dietary antioxidants, offering substantial health benefits to humans [7]. To further extend the portfolio of known bioactive compounds in coffee leaves, we applied bioinformatic methodologies with the aim of investigating metabolic pathways that may be constitutively expressed in *C. canephora* and *C. arabica* leaves. Our in silico analyses showed that *POLYPHENOL OXIDASES* (PPOs) and *DOPA DECARBOXYLASES* (*DDCs*) are expressed in leaves of both economically important coffee species.

PPOs are type III copper-containing metalloenzymes divided into three types: tyrosinases (TYRs, EC 1.14.18.1 and EC 1.10.3.1), CATECHOL OXIDASES (Cos, EC 1.10.3.1), and AURONE SYNTHASES (AUSs). AUSs are PPOs responsible for the synthesis of yellow pigments in the petals of various Asteraceae species [11]. PPOs are among the oldest enzymes known [12] and are widespread across all life kingdoms [13,14,15,16,17,18,19] with varying biological roles [16,20]. PPOs catalyze the oxidation of catechol to o-quinone in the presence of oxygen. The main difference between TYRs and COs is that the latter can only catalyze the oxidation of catechol (i.e., o-diphenol) to the corresponding o-quinone, whereas the former can catalyze both the monooxygenation of monophenols and the oxidation of catechols [21].

In animals and many microorganisms, PPOs are directly involved in the production of melanin pigments by oxidizing L-tyrosine (TYR) to L-DOPA (L-3,4-dihydroxyphenylalanine; levodopa) and other metabolites, ultimately producing dark-colored pigments [20,22]. Similarly, plant PPOs can produce dark-brownish compounds [23]. This browning process is the result of the oxidation of phenolics to quinones that are highly reactive intermediates involved in senescence, wounding, and response to pathogens [23]. After fruit harvest, the accumulation of these metabolites becomes evident. In many plant-derived foodstuffs, these reactions can reduce their nutritional quality and perceived commercial value [12,24,25].

It has been estimated that half of the world’s fruits and vegetable crops are lost due to postharvest deteriorative reactions [26]. This browning might be a side-effect of fundamental defense responses because, when PPOs are transcriptionally repressed in tomatoes (*Lycopersicon esculentum* L.), their susceptibility to the *P. syringae* is increased [23], whereas the overexpression of PPOs reduces the susceptibility to the same pathogen [27].

Apart from tyrosine, PPOs can accept diverse types of both monophenols and *o*-diphenols as substrates in different species and tissues. The monophenol substrates include, but are not limited to, 4-methylphenol [28], 4-propylphenol [29], 4-tert-butylphenol [30], 4-aminophenol [31], and p-tyrosol [29]. Similarly, *o*-diphenol substrates include 4-methylcatechol [28], L-3,4-dihydroxyphenylalanine (L-DOPA) [32], 4-tert-butylcatechol [32], 3,4-dihydroxyphenethylamine (dopamine) [29], caffeic acid [29], and 5-caffeoylquinic acid (5CQA; chlorogenic acid) [33]. In *C. arabica* leaves and the endosperm, the most efficient PPO substrate was found to be 5CQA, followed by 4-methilcatecol, caffeic acid, and catechol [33]. There is evidence that *C.arabica* cultivars in which leaves have higher concentration of 5CQA are more resistant to *Hemileia vastatrix,* a pathogen fungus to coffee plants and the causal agent of coffee leaf rust [6]. In addition, the PPO activity is higher in young coffee leaves and decreases with increasing leaf length and age [33,34].

In humans and other animals, L-DOPA is an important precursor of the neurotransmitter dopamine [35,36]. The depletion of dopamine in the human brain causes the neurodegenerative condition called Parkinson’s disease [37]. In humans, dopamine cannot cross the morphological barrier at the blood–brain interface while L-DOPA can [38]. Once L-DOPA enters the central nervous system, it is converted into dopamine by the enzyme DOPA decarboxylase (DDC; EC 4.1.1.28). Due to its capacity to traverse the blood–brain barrier and undergo metabolism by DDC, converting it into dopamine within the brain, L-DOPA serves as the standard treatment for Parkinson’s disease [35].

Plant DDCs are common enzymes that mediate numerous secondary reactions [39]. Nevertheless, the full extension of their biological function in plant growth and development remains unknown. DDC overexpression in apple trees increases both dopamine levels and salt tolerance [39]. This effect may be due to enhanced maintenance of ion homeostasis, which has been verified after dopamine treatment [40]. It has been reported that dopamine can promote nutrient uptake, transport, and distribution, in addition to promoting the downregulation of senescence-related genes [41]. In cucumber, it has been shown to meditate photosynthesis, as well as carbon and nitrogen metabolism, and reduce damage under nitrate stress [42]. Because dopamine can mediate important biological processes in plants, its production by DDC may be of relevance in coffee.

The full phenolic composition of *C. arabica* remains unknown [43,44]. To investigate potential metabolites, we evaluated the transcriptome profile of multiple *C. arabica* RNAseq leaves samples that are publicly available at the Sequence Read Archive (SRA) from the National Center for Biotechnology Information (NCBI). Then, we compared the expressed genes with metabolic pathways available at the Kyoto Encyclopedia of Genes and Genomes (KEGG) database [45].

We focused our attention on genes coding for the enzymes PPOs and DDC that are present in multiple copies in the genomes of *C. arabica* and *C. canephora.* We used the high-performance liquid chromatography with tandem mass spectrometry (HPLC–MS/MS) technique to show that L-DOPA is a phenolic metabolite that naturally occurs in *Coffea* leaves and fruits. To our knowledge, L-DOPA has not previously been documented as a naturally occurring phenolic compound in Coffea. Nevertheless, in vitro evidence suggests that a PPO extracted from the coffee endosperm can accept L-DOPA as a substrate, albeit with an activity 25.6 times lower than its preferred substrate, 5CQA [33]. The co-expression of PPO and DDC suggests that dopamine is also present in leaves of those species.

These results show that in silico analysis coupled with analytical chemistry techniques is a powerful combination of toolsets to allow the identification of compounds of economic and pharmacological importance in plants. In addition, these findings may provide additional base to the use of coffee leaves as a source of phenolic metabolites with medicinal, phytotherapeutic, and economic value.

## 2. Results

### 2.1. Exploratory Analyses of Metabolic Pathways Show That PPOs and DDCs Are Present in the Genomes of C. arabica and C. canephora

Our exploratory analyses of metabolic pathways in *Coffea* revealed important enzymes central for the survival of plants, such as those involved in carbon fixation in photosynthetic organs (Appendix A). Among the metabolites predicted to occur along the investigated pathways (Appendix A), we focused our attention on L-DOPA, which is a substrate of DDC and a product of PPO (Figure 1).

We found eight *PPOs* in the *C. arabica* genome, seven of which were encoded by the C. canephora sub-genome and one of which was encoded by the C. eugenioides sub-genome (Appendix A). In addition, we found three *PPOs* in the genome of *C. canephora* (Appendix A). All the *PPO* genes identified in the *C. arabica* genome were named *PPO.CAR* followed by a number from 1 to 8, whereas *C. canephora PPOs* genes were named *PPO.CCA* followed by a number from 1 to 3. The mean length of *Coffea* PPOs was 566 amino acids, ranging from 423 (PPO.CCA1) to 584 (PPO.CAR6). Their mean molecular weight was 63,336 g·mol^−1^ ranging from 48,056 g·mol^−1^ to 65,183 g·mol^−1^. Their mean isoelectric point (pI) was 6.28, ranging from 5.55 (PPO.CAR6) to 6.85 (PPO.CAR8). 

Regarding DDCs, we found six genomic loci in the *C. arabica* genome, four of which were encoded by the *C. eugenioides* sub-genome and two of which were encoded by the *C. canephora* sub-genome. In addition, we found three DDCs in the genome of *C. canephora* (Appendix A). All the *DDC* genes identified in the *C. arabica* were named *DDC.CAR* followed by a number from 1 to 6, whereas *C. canephora DDC* genes were called *DDC.CCA* followed by a number from 1 to 3. The mean length of *Coffea* DDCs was 508 amino acids, ranging from 480 (DDC.CAR6 and DDC.CCA3) to 537 (DDC.CAR2 and DDC.CCA2). Their mean molecular weight was 56,463 g·mol^−1^, ranging from 53,376 g·mol^−1^ to 59,696 g·mol^−1^. Their mean isoelectric point (pI) was 6.08, ranging from 5.77 (DDC.CAR6 and DDC.CA3) to 6.28 (DDC.CAR4).

The phylogeny inference for PPOs recapitulates the evolutionary pattern of angiosperms reported on the Angiosperm Phylogeny Website [46], as shown in Figure 2A. The three main clusters are representing the groups Asterids (containing the orders Solanales, Lamiales, and Gentianales), Rosids (containing the orders Rosales, Fabales, Fagales, Brassicales, and Malvales), and Monocots (containing the orders Arecales, Asparagales, Poales, and Zingiberales). Mostly all *Coffea* PPOs were clustered within the Asterid group, while a single PPO from *C. canephora* (PPO.CCA3) and two PPOs from *C. arabica* (PPO.CAR7 and PPO.CR8) were placed together with the Rosales *Trema orientale* and *Morus notabilis* (Figure 2A).

Seven DDC sequences were clustered within the Asterid group, five from *C. arabica* and two from *C. canephora* (Figure 2B). A cluster containing two DDCs, one from *C. canephora* (DDC.CCA3) and the other from *C. arabica-subgenome C. canephora* (DDC.CAR6), with more than 99% of identity, was placed outside of any of the main phylogenetic clusters, suggesting that these sequences are under evolutionary divergence after a gene duplication event that happened prior to the origin of the *C. arabica*.

Our expression analysis based on the identified *PPOs* and *DDCs* in the *C. arabica* genome suggests that all *PPOs* described here were expressed in fully expanded leaves of adult plants (Figure 2C). PPO.CAR1/2/3/4/5 (represented as PPO.CAR1-5 in the heatmap) were a group of highly similar PPOs with more than 98% of sequence identity at the protein level. In addition, these PPO.CAR1-5 displayed higher expression levels in *C. arabica* leaves when compared to the more distant CAR.PPO6 and the two PPOs clustered within the Rosid group CAR.PPO7/8. Regarding *DDCs*, only *DDC.CAR6* was found to be expressed in *C. arabica* leaves.

### 2.2. Chromatographic Analyses

The mean retention time for the L-DOPA standard solution was 3.51 ± 0.41 min (Appendix A). The selectivity was accessed by adding the standard solution to samples without L-DOPA. The fortified solutions were prepared by adding L-DOPA in two concentrations (50 and 100 µg·mL^−1^). Then, the fortified samples were compared to control samples without L-DOPA in the HPLC runs. By doing so, we were able to observe that the separation had no interference from the matrix in the identification of L-DOPA (Appendix A). The correlation coefficient (R^2^), detection limit (DL), quantification limit (QL), precision (CV), and accuracy (recovery) are presented in Table 1.

The measured R-squared (R^2^) value for L-DOPA in *Coffee* leaves was 0.99998 using HPLC. This is evidence of a strong linear correlation between L-DOPA concentration and peak area, as R^2^ values above 0.99 are widely accepted as indicators of linear relationships between chemical compound concentrations and their HPLC signal [48,49]. This can be observed in the calibration curve for the quantification of L-DOPA in human plasma using a similar approach which presented an R^2^ > 0.99 [50]. 

Using LC–MS/MS with multiple reaction monitoring (MRM), we were able to ascertain that the peak around 3.7 min of acquisition time was indeed L-DOPA, which is naturally synthesized by *C. arabica* fully expanded leaves and fruits (Table 2). This is because, at this specific time, we could identify all three reported transitions that are characteristic of L-DOPA [51,52,53]. In addition, L-DOPA was also found in *C. canephora* fully expanded leaves (Figure 3 and Appendix A). In both species, L-DOPA could not be verified in flowers because the LC–MS/MS signal was indistinguishable from the background noise. For this same reason, we could not verify the occurrence of L-DOPA in *C. canephora* fruits.

## 3. Discussion

Coffee leaves are byproducts of coffee culture that, in comparison to coffee beans, are mostly disregarded in studies concerning their chemical constituents [6,54,55]. However, the available studies reveal that coffee plants are rich in bioactive compounds such as echinoids, flavonoids, xanthones, and caffein [10]. Additionally, the potential of coffee and its derivatives, including tea, as excellent functional beverages has been recognized, with a history of traditional consumption spanning over 200 years in coffee-growing regions by local communities [54]. To further investigate the potential compounds in fully expanded coffee leaves, we applied a series of in silico analyses to infer metabolic pathways that may be active. Then, we selected a portion of the isoquinoline alkaloid biosynthesis pathway according to the KEGG [45] representation (map00950) which involves the enzymes PPO (EC 1.14.18.1, EC 1.10.3.1) and DDCs (EC 4.1.1.28—Figure 1). Lastly, we investigated the occurrence of the L-DOPA compound using HPLC and LC–MS/MS techniques.

### 3.1. Multiple PPO Copies Are Present in C. arabica and C. canephora Genomes

Multiple sequence alignment of PPOs showed that all coffee PPOs identified in this study were similar regarding their primary structure and presented the expected conserved domains (PF00264, PF12142, and PF12143) in the same order and positions in comparison to other functional plant PPOs. In addition, 10 out 11 coffee PPOs possessed a chloroplast transport signal peptide between amino acids 1 and 40 (Appendix A). It is well documented that many plant PPOs have this N-terminal domain containing a thylakoid transfer signal peptide to allow translocation through the chloroplast [56]. 

Only a PPO from *C. canephora*, named PPO.CCA1, did not present this signal peptide. It is not clear to us if this region was indeed absent in this *locus*, meaning that this specific PPO occurs outside the chloroplast, or if the predicted transcription start site was, in fact, upstream of the actual reported coordinates. Functional non-chloroplastic plant PPOs have been verified for poplar [57] and snapdragon [58] and are predicted to occur in monocots, such as rice and maize, and eudicots such as columbine [17]. The discovery of non-plastidic PPOs can help in the discovery of additional roles for PPOs in plants [17].

The PPO C-terminal domain PF12143 (whose PFAM description refers to as “unknown function domain; DUF_B2219”) is well conserved in all coffee PPOs. This domain is sometimes regarded to be a blocking-device for the active site via a placeholder residue. Nevertheless, this gatekeeping system is not functional in many plants, with *Malus domestica* and *S. lycopersicum* [29] being exceptions. On the other hand, this C-terminal domain is lacking in species such as *Vitis vinifera* [59]. It is believed that the C-terminal domain is a functional copper transporter system that works prior to a proteolytic cleavage [60].

The number of *PPOs* in plant genomes varies due to lineage-specific duplications, expansion, or loss [17]. *Arabidopsis,* a Malvidae, does not encode *PPOs,* whereas *Carica papaya*, also a Malvidae, encodes four *PPOs* [17]. Meanwhile, *Glycine max* and *Populus trichocarpa* genomes encode 11 *PPOs*. We found three *PPOs* in the *C. canephora* genome that may have originally arisen due to the ancestral whole-genome triplication event of eudicots [61,62]. Interestingly, we found eight *PPOs* in the *C. arabica* genome, with seven of them being located in the sub-genome derived from the ancestral parent *C. canephora* and one from the ancestral parent *C. eugenioides.*

The reason for the disparity in the number of PPOs between the *C. arabica* sub-genomes and the genome of current *C. canephora* plants remains unclear. This difference could be attributed to events that took place prior to the origin of *C. arabica*, possibly within the respective lineages of its parent progenitors. In this manner, it is plausible that the ancestors of *C. canephora* retained duplicated PPO loci as a result of dosage sensitivity gene balancing [63]. Later, this trait was kept in *C. arabica.*

### 3.2. Multiple PPO Copies Are Expressed in C. arabica Leaves

RNAseq expression analysis showed that fully expanded *C. arabica* leaves constitutively transcribed *PPOs* (Figure 2C). However, due to the high similarity of PPOs at the coding sequence level, the determination of which *PPO* loci were producing transcripts is not possible with current similarity-based bioinformatic approaches. For example, PPO.CAR1 and PPO.CAR2 were 100% identical throughout their length of 1734 nucleotides. In addition, PPO.CAR3, 4 and 5, all having the same length of 1734 nucleotides, were more than 98% similar to PPO.CAR1/2. For that reason, when transcript fragments from NGS were mapped to them, the algorithms could not discern if all loci were being expressed or just a subset of them.

These five remarkably similar PPOs were distinct loci located on chromosome 5, originating from the parental ancestral *C. canephora* (Appendix A). They were all single exon genes positioned closely together, strongly indicating their recent emergence from a local gene duplication event. *PPO.CAR6* was also located on chromosome 5 originating from the parental ancestral *C. canephora;* its protein had a similarity of 96.75% to a PPO in chromosome 5 of the currently living *C. canephora* plants and was also expressed in fully expanded leaves (Figure 2C). Lastly, CAR.PPO7 and CAR.PPO8 were 98.79% similar to each other but only ~50% similar to other PPOs in *C. arabica.* This may be the reason why they were clustered outside the Rosid group in Figure 2A. The PPO.CAR7 locus was located on chromosome 2 and originated from the parental ancestral *C. canephora*, whereas PPO.CAR8 was the only locus encoded in the sub-genome originating from the ancestral parent *C. eugenioides*. Additionally, PPO.CAR8 was also situated on chromosome 2 of its sub-genome.

### 3.3. Multiple DDC Copies Are Present in C. arabica and C. canephora Genomes but Only One Copy Is Expressed in C. arabica Fully Expanded Leaves

Multiple sequence alignment of DDCs showed that all coffee DDCs were similar regarding their primary structure and presented the expected pyridoxal-dependent decarboxylase conserved domain (Pyridoxal_deC—PF00282) in the middle section of these sequences (Appendix A). This domain occupied approximately 60% to 70% of the length of all DDCs evaluated, and it was the only conserved domain characteristic of DDCs.

In *C. arabica*, we found four DDCs encoded by the *C. eugenioides* sub-genome (two on chromosome 11e, one on chromosome 1e, and another on chromosome 9e) and two encoded by the *C. canephora* sub-genome (both close to each other on chromosome 9c). Similarly, the genome of the currently living *C. canephora* presented three DDCs, with one located on chromosome 1 and two organized in tandem on chromosome 9. Interestingly, all but two coffee DDCs, one from *C. arabica* DDC.CAR6 and other from *C. canephora* DDC.CCA3, were single-exon genes. These multi-exon DDCs were on chromosome 1 of their respective genomes, and they were the most divergent coffee DDCs in our phylogenetic analysis (Figure 2B and Appendix A).

The identity among the multi-exon DDCs, DDC.CCA3, and DDC.CAR6 was 99% throughout their protein length of 480 amino acids. DDC.CAR6 from *C. arabica* was on the sense (+) strand on its genome, whereas DDC.CCA3 from *C. canephora* genome was on the antisense (−) strand. In addition, these multi-exon DDCs were shorter and presented only ~55% of identity to other DDCs in coffee. The expression of DDC in *C. arabica* leaves was only verified for DDC.CAR6 (Figure 2C). It is not clear if this DDC preferentially catalyzes the conversion of L-tyrosine to tyramine or L-DOPA to dopamine. It is also possible that these enzymes catalyze different reactions. Some DDCs seem to be exclusively found in coffee because our phylogenetic inferences clustered the DDC CAR6 and CCA3 outside of any higher plant group evaluated in this study (Figure 2B). Lastly, the apparently silenced DDC 1–5 in leaves may be active in other leaves under a different set of environmental conditions.

### 3.4. Chromatography Analyses Confirmed the Presence of L-DOPA in Coffee Leaves and Fruits

The technical DL and QL for our chromatography analysis were 0.81 and 2.73 µg·mL^−1^, respectively, for L-DOPA in coffee extracts. Both values were above the reports for human plasma, DL of 0.025 and QL of 0.1 μg/mL [50]. On the other hand, Pavón-Pérez et al. (2019) [64] reported a DL of 0.01 mg·L^−1^ and a QL of 0.05 mg·L^−1^ using LC–MS/MS on *Vicia faba* extracts. Meanwhile, the reported QL for L-DOPA in rat plasma was 25.0 ng/mL [52], which is higher than our findings. The differences found in these parameters may arise from differences in the studied matrices and in the chromatographic conditions, such as in the equipment and/or methodologies used for detection and quantification.

Our recovery assays to determine the accuracy of the technique returned average values ranging from 81% to 104% (with CV ranging from 0.38% to 1.11%). Those values are within the analytical acceptance range of 70% to 120% with ±20% precision (CV) [65]. On the basis of the results found in this study regarding the recovery values for the L-DOPA compound, we propose that the applied method showed satisfactorily good recovery. Our findings are in accordance with other studies that reported recovery yields of 94% to 117% (relative standard deviation of ≤5.66) [64], 98% to 106% (CV ≤ 15%) [50]. In addition, our recovery was higher than the values reported for the quantitation of L-DOPA in rat plasma by HPLC–UV/Vis in which values ranged from 46.5% to 50.1% (CV ≤ 10.3%) [52]. Our methodology presented a CV ranging from 0.38% to 1.11%. In this work, the CV value was below 5%, which is the precision lower limit for compounds found in low concentrations in biological extracts [48,49,65]. 

Although the samples of fully expanded leaves from *C. arabica* and *C. canephora* and fruits from *C. arabica* presented analytical signals below 50 μa, it was possible to confirm the presence of L-DOPA in those organs (Figure 3). In addition, the presence of higher-intensity interference in the MRM transition 198 > 152 is notable since the choice of mass transitions in MRM mode allows analysis in qualitative (confirmatory) mode. Lastly, this specific transition may suggest the decarboxylation of L-DOPA at C9, resulting in the production of dopamine.

## 4. Material and Methods

### 4.1. Identification of Enzyme-Coding Genes in Coffee Genome and Inference of Metabolic Pathways

A total of 25,605 predicted protein sequences in *C. canephora* were downloaded from the Coffee genome hub [61]. Then, they were analyzed with the blast2GO [66] suite to search for potential enzymes and their respective enzyme codes (ECs). We used the resulting list of 1141 nonredundant ECs through the online KEGG mapper tool [45] to find metabolic pathways that were possibly active in *Coffea.* As part of our comprehensive search, we incorporated the following reference KEGG pathway map databases: carbohydrate metabolism (1.1), energy metabolism (1.2), lipid metabolism (1.3), amino-acid metabolism (1.5), metabolism of cofactors and vitamins (1.8), metabolism of terpenoids and polyketides (1.9), and biosynthesis of secondary metabolites (1.10).

### 4.2. Characterization of PPO- and DDC-Coding Genes in Coffea

The genome sequence of the *C. arabica* Caturra-red cultivar was retrieved from the NCBI under BioProject accession PRJNA506972 [67]. Then, we predicted protein coding genes using AUGUSTUS v. 3.3.3 [68]. The annotation of protein coding genes was performed using blast2GO [66]. Sequences with the enzymatic code for PPOs (EC 1.14.18.1 and EC 1.10.3.1) and DDC (EC 4.1.1.28) where selected from both *C. arabica* and *C. canephora*, and conserved domain analyses were performed using hidden Markov models by aligning the selected protein sequences against the Pfam domain database v35 [69] with the HMMER software v3.3.2 [69]. Finally, the respective coding sequences for each putative PPO and DDC were aligned to the NCBI nonredundant (nr) protein database using blastx v 2.12.0+ [70].

We considered coffee PPOs those protein sequences (1) that presented significant hits to the three typical plant PPO domains in the following amino-carboxyl order: tyrosinase (PF00264), polyphenol oxidase middle domain (PPO1_DWL; PF12142), and PPO1_KFDV (PF12143), (2) with at least 70% of length coverage and 50% of identity to the 3D-chistolograph verified PPO structure from *Ipomoea batatas* (UniProtKB/Swiss-Prot: Q9MB14.2) [71], and (3) with the five blastx top-hits of known plant PPOs. In addition, we considered the coffee DDC homologs those protein sequences (1) that presented significant hits to the pyridoxal-dependent decarboxylase conserved domain (PF00282.22), (2) with at least 70% of coverage and 50% of identity to the curated DDC from *Papaver somniferum* (UniProtKB/Swiss-Prot: P54768), and (3) with the five blastx top-hits of known plants DDC, not including the highly similar proteins L-tryptophan decarboxylase (TDC2-like). Then, their physicochemical properties (length of amino-acid sequence, molecular weight, and isoelectric point) were determined with the ExPASy Proteomics tool (https://web.expasy.org/protparam/ accessed on 27 August 2022).

### 4.3. Phylogenetic Analysis

Representative protein sequences for PPOs and DDC were retrieved from NCBI’s nr database for the following taxa: Amborellales, Arecales, Asparagales, Asterales, Brassicales, Cannabaceae, Cucurbitales, Fabales, Gentianales, Ginkgoales, Lamiales, Liliales, Lycophytes, Malpighiales, Malvaceae, Poales, Ranunculales, Rosales, Solanales, Vitales, and Zingiberales. A tyrosinase from *Homo sapiens* (GenBank accession AAA61244.1) was used as an outgroup for plant PPOs, and a human DDC (NCBI accession: NP_000781.2) was used as an outgroup for plant DDC. The selected plant species represents diverse phylogenetic groups of higher plants. To this subset of PPOs or DDCs, we added the respective homologs from *C. arabica* and *C. canephora.*

The multiple protein sequence alignment was performed with MAFFT v. 7.505 using the iterative refinement method incorporating global pairwise alignment information (G-INS-i) [72,73]. For inferring phylogenetic relationships, coffee sequences with more than 98% of identity at the protein level were collapsed into a single representative sequence. Phylogenetic trees were inferred with PHYLIP [74] v. 3.696 with 1000 bootstrap replicates, using the Jones–Taylor–Thornton substitution model [75] and neighbor-joining clustering method [76]. The consensus tree was chosen by the majority rule and drawn using the Interactive Tree of Life (iTOL v. 6.5.8) webtool [47]. Transfer signal peptides were inferred with the online tool LOCALIZER v. 1.0.4 [77]. 

### 4.4. Expression Evaluation of PPOs and DDCs

To identify expressed PPOs and DDCs in *C. arabica* leaves, we downloaded paired-end RNAseq libraries available at the SRA of the NCBI under bioproject ID PRJNA851465 [78]. In summary, the experiment was conducted in Brazilian farms of two cities (Pirapora and Varginha) during two harvest times (April and October) and with two *C. arabica* cultivars (Acauã and Catuaí Vermelho IAC 144). After quality assessment steps, the RNAseq reads were mapped to the *C. arabica* reference genome (BioProject accession PRJNA506972 [67]) using the STAR aligner v. 2.7.8 [79]. Then, fragments mapped to gene exons were quantified with the HTseq-count script [80] and analyzed with edgeR [81]; an expression-based heatmap was produced with the heatmap.2 function from gplot package [82]. 

### 4.5. Extraction of L-DOPA from C. arabica Leaves

The extraction procedure was based on a sustainable, simple, and robust method for L-DOPA extraction recently developed for *Vicia faba* [83]. We collected *C. arabica* leaves and immediately macerated them with liquid nitrogen until a fine and homogeneous powder was produced. Samples of 200 mg were collected in 15 mL tubes with 5 mL of acetic acid 0.1%. Then, we homogenized samples for 20 min with a magnetic shaker and subsequently centrifuged at 13,000 rpm for 10 min at room temperature (±25 °C). Next, we collected the supernatants, and a second extraction step was performed with the remaining biomass. Lastly, we mixed and filtered both supernatants in a membrane and immediately submitted them to chromatographic analysis.

### 4.6. Liquid Chromatographic Analysis and Validation Parameters

The analyses were performed at the Brazilian National Institute of Coffee Science and Technology (Instituto Nacional de Ciência e Tecnologia do Café; INCT-Café) at the Federal University of Lavras (Universidade Federal de Lavras; UFLA). The liquid chromatographic runs were performed with a Shimadzu HPLC equipment composed of a high-pressure quaternary pump model LC-20AT, a degasser DGU-20A5, an interface CBM-20A, an automatic injector SIL-20A-HT, and a UV/Vis detector SPD-20A. The used column was a Zorbax Eclipse XDB-C18 (4.6 × 250 mm, 5 µm) connected to an XDB-C18 pre-column (4.6 × 12.5 mm, 5 µm).

The L-DOPA analysis was performed using the methodology proposed by Elbarbry et al. (2019) [50] with modifications. L-DOPA standard was purchased from Sigma-Aldrich (St. Louis, MO, USA). Mobile-phase chemicals were all of HPLC analytical grade: methanol (Merck), glacial acetic acid (J.T.Baker), and type I water from a Milli-Q system.

We used the external standardization method to apply quantification procedures. For the analytical curves, we diluted a stock solution with the L-DOPA standard in perchloric acid (1000 µg·mL^−1^). From that stock solution, we prepared the analytical curve by varying the concentration from 0.1 to 200 µg·mL^−1^. The selected mobile phase for the compound elution was acetic acid 1% in water (Solvent A) and methanol (Solvent B) at a ratio of 95:5 (*v*/*v*) and a flow rate of 1.0 mL·min^−1^. We eluted samples and standards in isocratic mode at 30 °C in the column oven. The used light wavelength was 282 nm, and the injection volume was 20 µL. 

We filtered the biological samples and standard solutions in a 0.45 µm polyethylene membrane (Millipore) and injected them directly into the chromatographic system. The injections of the standards and biological samples were performed in triplicate, with the analyte identity confirmed by the retention time and the peak profile of the sample compared to that of the standard solution.

To ensure the analytical quality of the results, we evaluated multiple parameters such as selectivity, linearity, detection limit (DL), quantification limit (QL), precision (in terms of coefficient of variation, CV), and accuracy (recovery). All the procedures required to evaluate those parameters were performed to guarantee the standardization of the method [48,49,65]. Firstly, we evaluated the selectivity by adding to a pool of samples in different quantities of the L-DOPA standard. Then, we evaluated the linearity by inferring the linear regression equation and Its respective correlation coefficient (R^2^). An R^2^ greater than 0.99 was considered as evidence of an ideal fit of the data to the model.

To verify the ascertainment of detection (DL) and quantification (QL) limits, we considered the parameters related to the selectivity linear regression curve. To this end, we applied the following equations: DL = 3 × (*s/S*) and QL =10 × (*s/S*), where *s* is the standard deviation estimate of the linear regression model, and *S* is its slope.

The precision was calculated using the intermediate precision method. To do so, we repeated the HPLC analysis for 5 days by evaluating the readings of standard solutions with three known concentrations (1.0, 50.0, and 100.0 µg·mL^−1^). At the end, the coefficient of variation (CV), expressed as a percentage, was calculated with the function CV = (*s*/DMC) × 100, where *s* is the estimated standard deviation, and *DMC* is the determined mean concentration.

Lastly, we evaluated the accuracy by running recovery assays using three random samples fortified with standard solutions at three concentration levels (1.0, 50.0, and 100.0 µg mL^−1^). The recovery, expressed as a percentage of L-DOPA, was determined using the following equation: recovery = [(measured concentration)/(expected concentration)] × 100.

### 4.7. LC–MS/MS for Qualitative Analyses

To verify the occurrence of L-DOPA in *Coffea,* sample extracts in triplicate from *C. arabica* and *C. canephora* leaves, flowers, and fruits were analyzed by LC–MS/MS. Those analyses were performed in an Agilent Technologies system consisting of a binary pump, a degassing unit, a G4226A autosampler, a column oven, and a triple-quadrupole mass spectrometer (QqQ G6420A). The system was controlled by MassHunter Workstation Software (Version B.08.00). The separation was carried out on Zorbax Eclipse XDB-C18, 4.6 × 250 mm × 5 µm, thermostated at 30 °C, using a mobile phase composed of acetic acid 1% in water (Solvent A) and methanol (Solvent B) at a ratio of 95:5 (*v*/*v*) and a flow rate of 1.0 mL·min^−1^. Full scan spectra were acquired from *m*/*z* 10 to 500. Identification of L-DOPA was performed in multiple reaction monitoring (MRM) mode, detecting the following transitions: *m*/*z* 198 → *m*/*z* 152, *m*/*z* 198 → *m*/*z* 107, and *m*/*z* 198 → *m*/*z* 135 [51,52,53].

## 5. Conclusions

Our in silico analysis revealed that the *C. arabica* genome contains multiple copies of PPOs and DDCs, with some of these genes being expressed in fully expanded leaves. Notably, L-DOPA, one of the products of the PPO enzyme, was detected in both *C. arabica* and *C. canephora* leaves. This finding suggests that L-DOPA in fully expanded leaves could potentially play a role in promoting defensive mechanisms against pathogens, possibly involving its conversion to dopamine by DDCs. The presence of dopamine as a naturally occurring metabolite in coffee leaves may be indicative of a mechanism to alleviate nutrient deficiency-induced stresses [42].

Future research endeavors are needed to comprehensively unveil the significance of L-DOPA, PPOs, and DDCs in the context of coffee. It is possible that younger leaves could exhibit elevated L-DOPA concentrations, considering the potential higher PPO activity found in those leaves. Further investigations will shed light on the precise roles and importance of these components in coffee plants [33,34]. Additionally, different coffee varieties or wild *C.* species may be an enhanced source of L-DOPA. This work advances toward the purpose of using coffee leaves as a source of compounds with nutraceutical importance. Lastly, we demonstrated that in silico analysis is an effective tool to predict metabolic pathways, whose intermediate compounds can be verified using in vitro approaches such HPLC and related techniques. Employing the same in silico approach, we discovered additional potential pathways that could serve as valuable guides for in vitro studies, helping to uncover essential metabolites in coffee.

## Figures and Tables

**Figure 1 ijms-24-12466-f001:**
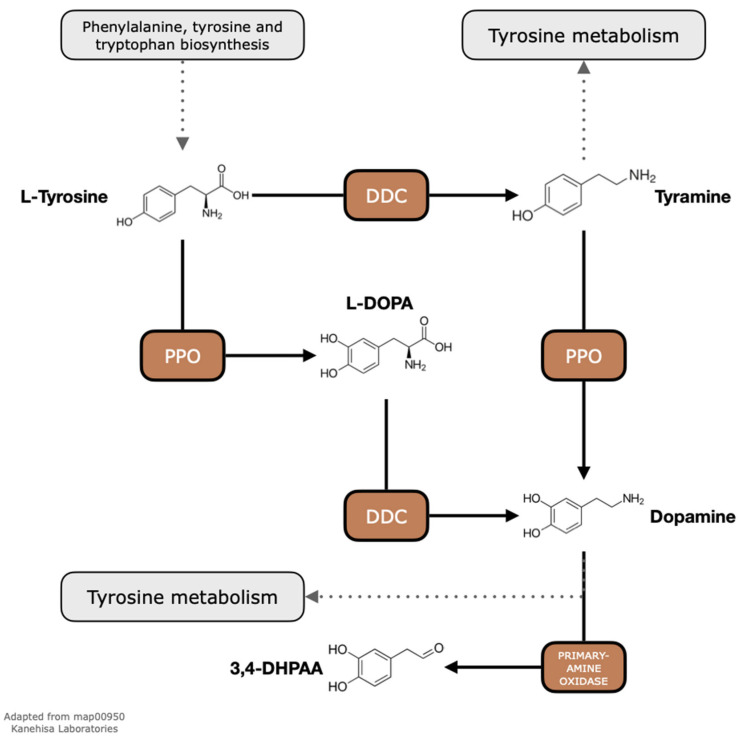
Representation of part of the isoquinoline alkaloid biosynthesis pathway according to the KEGG map00950. Here, we show only the enzymes whose genes could be identified in the coffee genome. *POLYPHENOL OXIDASE* (PPOs; EC:1.14.18.1 and EC:1.10.3.1) and *DOPA DECARBOXYLASE* (DDC; EC:4.1.1.28) can both accept the amino acid L-tyrosine as a substrate, producing L-DOPA (L-3,4-dihydroxyphenylalanine) and tyramine, respectively. Then, L-DOPA becomes an intermediate metabolite that can be used as a substrate to DDC to produce dopamine. Alternatively, dopamine can also be produced from a tyramine substrate in a PPO enzyme. Among other factors, the final concentration of L-DOPA and dopamine will depend on the downstream pathways such as the tyrosine metabolism or the synthesis of 3,4-DHPAA (3,4-dihydroxyphenylacetaldehyde) by a primary-amine oxidase (EC:1.4.3.21).

**Figure 2 ijms-24-12466-f002:**
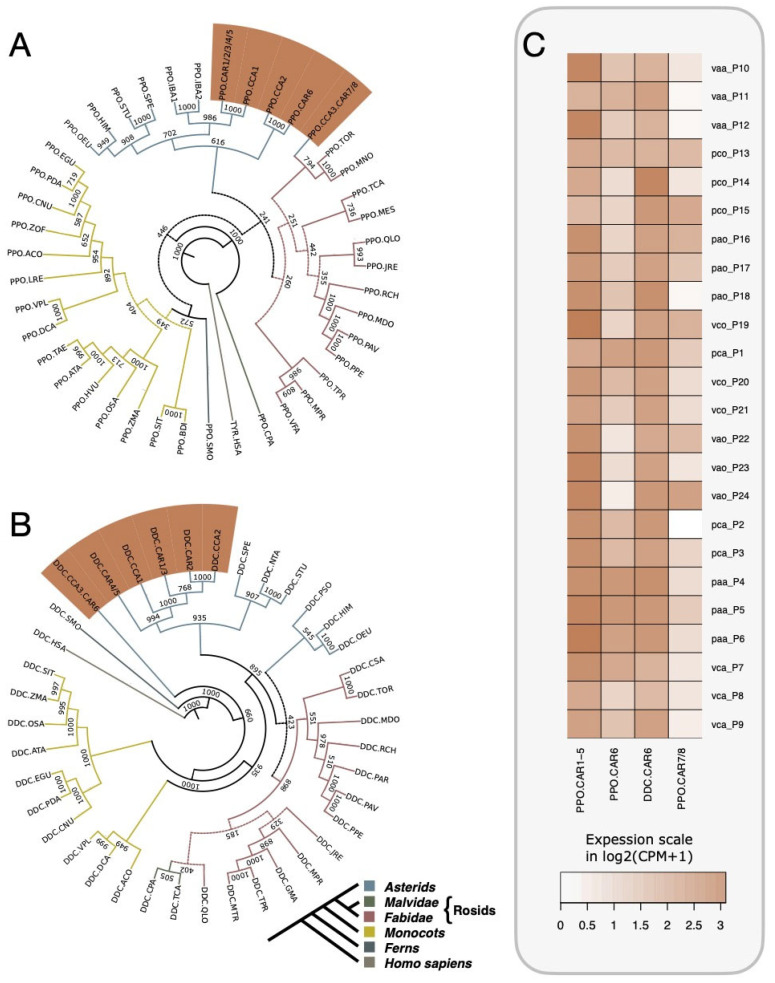
Consensus tree using the neighbor-joining clustering method depicting the evolutionary relationships of *Coffea arabica* (CAR) and *Coffea canephora* (CCA) polyphenol oxidases (PPOs) and DOPA decarboxylases (DDCs), as well as the expression profile of their respective genes in *C. arabica* leaves. (**A**) Phylogenetic tree of PPOs; five PPOs in *C. arabica* (PPO.CAR 1–5) with identity above 98% were collapsed during phylogeny inference. PPO.CAR1/2/3/4/5 were clustered together with other PPOs from *C. arabica, C. canephora*, and other members of the Asterid group. One PPO from *C. canephora* (PPO.CCA3) and two PPOs from *C. arabica* (PPO.CAR7 and PPO.CAR8) were clustered into the Rosid group, suggesting that they are under functional diversification. (**B**) Phylogenetic tree of DDCs; five DDCs from *C. arabica* (DDC.CAR 1 to 5) clustered with two from *C. canephora* (DDC.CCA1 and 2), as well as other members of Asterids. However, two highly similar DDCs, one from *C. canephora* (DDC.CCA3) and the other from *C. arabica* (DDC.CAR6), were clustered outside of any other flowering plant group, suggesting a diversification of this gene in *Coffea.* (**C**) Heatmap representation of the expressed *PPOs* and *DDCs* using RNAseq data from *C. arabica* fully expanded leaves in a field experiment with two cultivars (Acauã or Catuaí) and two harvest times (April or October), grown in farms of two Brazilian cities (Pirapora or Varginha). Expression values are normalized as counts per million (CPM) and represented on a log_2_(CPM + 1) scale. Each line in the heatmap represents a sequenced library from eight biological samples: Varginha, Catuaí, October (vco); Varginha, Catuaí, April (vca); Varginha, Acauã, October (vao); Varginha, Acauã, April (vaa); Pirapora, Catuaí, October (pco); Pirapora, Catuaí, April (pca); Pirapora, Acauã, October (pao); Pirapora, Acua, April (paa). Each biosample consists of three biological replicates. *PPO.CAR1-5* (representing five highly similar loci with identity above 98%) was constitutively expressed in all analyzed RNAseq samples of leaves, while PPO.CAR6 and the divergent PPO.CAR7/8 were less expressed. Regarding *DDCs*, only *DDC.CAR6* was found to be expressed in *C. arabica* leaves. In both phylogenetic trees (**A**,**B**), node numbers correspond to the sum of occurrences of pairs of groups or individual sequences that clustered together in a total of 1000 bootstraps; dashed lines represent nodes in which group pairs were clustered together in less than 500 (50%) of the bootstraps. The selected species and their respective codes are *Ananas comosus* (ACO), *Aegilops tauschii* (ATA), *Brachypodium distachyon* (BDI), *C. arabica* (CAR), *C. canephora* (CCA), *Cocos nucifera* (CNU), *Carica papaya* (CPA), *Cannabis sativa* (CSA), *Dendrobium catenatum* (DCA), *Elaeis guineensis* (EGU), *Glycine max* (GMA), *Handroanthus impetiginosus* (HIM), *Homo sapiens* (HSA), *Hordeum vulgare* (HVU), *Ipomoea batatas* (IBA), *Juglans regia* (JRE), *Lilium regale* (LRE), *Malus domestica* (MDO), *Manihot esculenta* (MES), *Morus notabilis* (MNO), *Mucuna pruriens* (MPR), *Medicago truncatula* (MTR), *Nicotiana tabacum* (NTA), *Olea europaea* (OEU), *Oryza sativa* (OSA), *Prunus armeniaca* (PAR), *Prunus avium* (PAV), *Phoenix dactylifera* (PDA), *Prunus persica* (PPE), *Papaver somniferum* (PSO), *Quercus lobata* (QLO), *Rosa chinensis* (RCH), *Setaria italica* (SIT), *Selaginella moellendorffii* (SMO), *Solanum pennellii* (SPE), *Solanum tuberosum* (STU), *Triticum aestivum* (TAE), *Theobroma cacao* (TCA), *Trema orientale* (TOR), *Trifolium pratense* (TPR), *Vicia faba* (VFA), *Vanilla planifolia* (VPL), *Zea mays* (ZMA), and *Zingiber officinale* (ZOF). GenBank or UniProtKB/Swiss-Prot IDs are available in Appendix A (PPOs) and Appendix A (DDCs). The small phylogenetic tree at the bottom is based on the Angiosperm Phylogeny Website [47], and colors represent specific phylogenetic groups; homologous *Homo sapiens* proteins were add as the outgroup.

**Figure 3 ijms-24-12466-f003:**
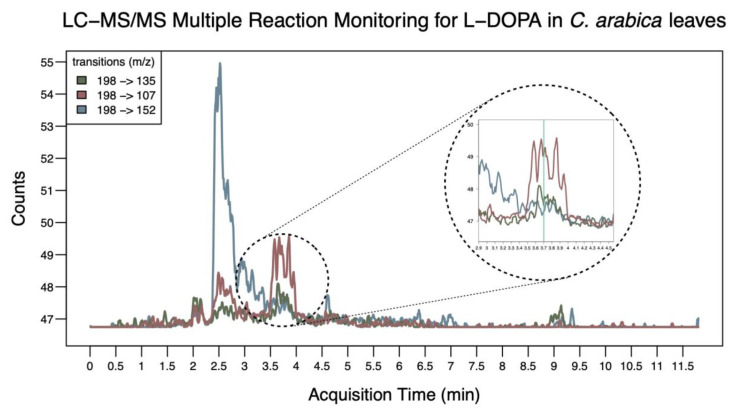
Sequential mass spectrometry with multiple reaction monitoring (MRM) profile. In the expected retention time (tR ≈ 3.7 min, highlighted circular section), the presence of the three specified transitions was verified, indicating the presence of L-DOPA molecule in the *Coffea arabica* leaf sample. In addition, the presence of higher-intensity interference in the transition 198 > 152 (blue) is notable, which may suggest the decarboxylation of L-DOPA in carbon C9, resulting in the production of dopamine.

**Table 1 ijms-24-12466-t001:** Analytical parameter for the method standardization.

Parameter	L-DOPA
B (linear coefficient)	1694.5
A (angular coefficient)	177.4
R^2^	0.99998
DL (µg·mL^−1^)	0.81
QL (µg·mL^−1^)	2.73
Recovery (%)	81 to 104
CV (%)	0.38 to 1.11

**Table 2 ijms-24-12466-t002:** LC–MS/MS analysis results from samples of leaves, flowers, and fruits extracted from *Coffea arabica* and *Coffea canephora*. Samples with readings above background noise were verified for the presence of L-DOPA.

Species	Tissue	Sample ID	Counts	Signal Above Noise Level
*Coffea arabica*	Leaves	1	48	Yes
		2	48	Yes
		3	48	Yes
	Flowers	4	47	No
		5	47	No
		6	47	No
	Fruits	7	48.5	Yes
		8	49	Yes
		9	49.5	Yes
*Coffea canephora*	Leaves	10	48.5	Yes
		11	48	Yes
		12	49	Yes
	Flowers	13	48	No
		14	48	No
		15	48	No
	Fruits	16	47	No
		17	48	No
		18	48	No

## Data Availability

The codes are available at https://github.com/thalescherubino/thesisChapter2/. GenBank IDs for homologous proteins are available in Appendix A.

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
