# Peer review of "Metabolic Pathway Reconstruction Indicates the Presence of Important Medicinal Compounds in Coffea Such as L-DOPA"

_ijms, 2023, doi:10.3390/ijms241512466_

Round 1

Reviewer 1 Report

In this work, the authors have analyzed the metabolic pathways of Coffea arabica and Coffea canephora by exploring RNAseq data available in open databases. Subsequently, they identified two enzymes associated with L-DOPA metabolism- DOPA DECARBOXILASES and POLYPHENOL OXIDASES. Finally, they demonstrated the presence of L-DOPA in coffea leaves and fruits using HPLC and LCMS. The claim is novel, and of interest, however, there is a scope for improvement both from the writing and experimental perspectives.

My comments are as follows:

Major:

I have two major comments: 1. To validate the claims that DDCs and PPOs, as reported in section 3.1, are responsible for L-DOPA synthesis, they need to show whether the changes in their expression levels correlate with the L-DOPA production. Since the authors already have established an HPLC setup for L-DOPA detection, I suggest, they can either transiently silence / overexpress these enzymes in coffea leaves and measure the corresponding changes in L-DOPA level. Or they can choose those samples like Coffea arabica- flowers, Coffea canephora- flowers and fruits, where L-DOPA was not produced, and show the levels of DDCs and PPOs therein by qPCR, if it correlates with the L-DOPA levels.

2. This study confirms that L-DOPA is produced in both Coffea arabica- and Coffea canephora leaves, however, whether it would be of any use, will be determined by the quantities they are present therein. Therefore, to give the readers a perspective on that front the authors should include the amount of L-DOPA that is produced/recovered per unit weight of coffea leaves.

Minor:

The text has many editing errors and needs thorough corrections. I have pointed out a few here.

1. Abstract (Line 20-23) does not read well, (Line 29-33) seems out of place. Please rephrase the abstract.

2. For DDCs throughout the text it advised to change ‘DESCARBOXILASES’ to the English form ‘DECARBOXYLASES’.

3. Introduction is too lengthy and needs to be concise.

4. Line 380, define ‘samples’ here.

5. Table 1, define ‘B’ and ‘A’ here.

6. Line 548, correct ‘DCC’.

The text has many editing errors and needs thorough corrections. I have pointed out a few in my minor comments.

Reviewer 2 Report

Dear Authors,

I recommend your article  to be published in ithe present form.

Reviewer 3 Report

The scientific paper appears to be a thorough, methodical, and meticulous exploration of the presence and roles of POLYPHENOL OXIDASES (PPOs) and DOPA DESCARBOXILASES (DDCs) in Coffea arabica and Coffea canephora leaves. The research benefits from comprehensive in silico analyses and an extensive review of existing literature. The study reveals the existence of L-DOPA in Coffea leaves, a compound known for its medical importance, especially in treating Parkinson's disease, and suggests a potential defense mechanism against pathogens.
However, the paper seems to indicate the need for future studies to fully elucidate the roles of L-DOPA, PPOs, and DDCs in coffee, suggesting a potential lack of experimental data within this specific research. Also, the researchers' intention to use coffee leaves as a source of medically important compounds adds a potentially beneficial economic and health-related aspect to their study. This piece of research is a promising contribution to the understanding of coffee physiology, its potential health benefits, and offers future research avenues.

Round 2

Reviewer 1 Report

OK from my side.